# How Multimodal LLMs Solve Image Tasks: A Lens on Visual Grounding, Task Reasoning, and Answer Decoding

**Zhuoran Yu**
University of Wisconsin–Madison
zhuoran.yu@wisc.edu

**Yong Jae Lee**
University of Wisconsin–Madison
yongjaelee@cs.wisc.edu

## Abstract

Multimodal Large Language Models (MLLMs) have demonstrated strong performance across a wide range of vision-language tasks, yet their internal processing dynamics remain underexplored. In this work, we introduce a probing framework to systematically analyze how MLLMs process visual and textual inputs across layers. We train linear classifiers to predict fine-grained visual categories (e.g., dog breeds) from token embeddings extracted at each layer, using a standardized anchor question. To uncover the functional roles of different layers, we evaluate these probes under three types of controlled prompt variations: (1) lexical variants that test sensitivity to surface-level changes, (2) semantic negation variants that flip the expected answer by modifying the visual concept in the prompt, and (3) output format variants that preserve reasoning but alter the answer format. Applying our framework to LLaVA-1.5, LLaVA-Next-LLaMA-3, and Qwen2-VL, we identify a consistent stage-wise structure in which early layers perform visual grounding, middle layers support lexical integration and semantic reasoning, and final layers prepare task-specific outputs. We further show that while the overall stage-wise structure remains stable across variations in visual tokenization, instruction tuning data, and pretraining corpus, the specific layer allocation to each stage shifts notably with changes in the base LLM architecture. Our findings provide a unified perspective on the layer-wise organization of MLLMs and offer a lightweight, model-agnostic approach for analyzing multimodal representation dynamics.

## 1 Introduction

Multimodal Large Language Models (MLLMs) (Liu et al., 2024b; Bai et al., 2023; Wang et al., 2024; Zhu et al., 2023; Li et al., 2023) have recently demonstrated remarkable capabilities in integrating visual and textual inputs to perform tasks such as image captioning (Chen et al., 2015; Agrawal et al., 2019), visual question answering (Goyal et al., 2017; Marino et al., 2019; Hudson & Manning, 2019), and open-ended instruction following (Hendrycks et al., 2021; Zhong et al., 2023). While these models exhibit strong performance across benchmarks, their internal processing dynamics—particularly how visual and linguistic information interact across layers—are not yet fully understood. Understanding this internal structure is important not only for interpretability but also for gaining deeper insights into model behavior and limitations.

Several recent studies have begun to explore the internal mechanisms of MLLMs using tools such as causal tracing (Basu et al., 2024; Neo et al., 2024), logit lens (Huo et al., 2024), and concept-level perturbation (Golovanevsky et al., 2024). These methods provide valuable insights into token-level attribution, attention behavior, and concept encoding. However, they often require custom instrumentation or focus on isolated components, making it difficult to obtain a high-level understanding of how information flows through the model.

In this work, we introduce a probing-based framework that examines how prompt variations affect layer-wise representations during multimodal inference. We treat each transformer

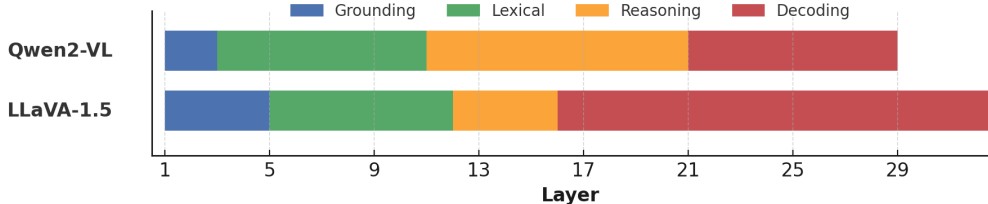

Figure 1: **Layer-wise stage comparison between LLaVA-1.5 and Qwen2-VL.** Each colored segment corresponds to a distinct functional phase identified by our probing analysis: **Grounding** (visual input encoding), **Lexical Integration** (alignment between image and prompt phrasing), **Semantic Reasoning** (internal answer formulation), and **Decoding** (output generation). While both models exhibit the same stage-wise structure, Qwen2-VL allocates fewer layers to grounding and more to reasoning, reflecting architectural differences in how multimodal understanding is distributed across layers.

layer $l$ in the language model as a feature extractor, and train a linear classifier $f^{(l)} : \mathbb{R}^d \to [1, \ldots, N]$ to predict visual class labels from the final token representation $\mathbf{h}^{(l)} \in \mathbb{R}^d$. Training is conducted on a set of $N$ image classes paired with a fixed textual prompt—referred to as the *anchor prompt*—which is held constant across all training instances.

At test time, we evaluate the same probes on a held-out set of images from the same $N$ classes, each paired with systematically perturbed versions of the anchor prompt. Each prompt variant is designed to target a specific aspect of the input—such as lexical form, semantic content, or output format—while preserving the correct visual label. The core assumption is that different layers of the transformer are specialized for different types of computation. Therefore, the extent to which probe accuracy is affected by each variant reflects the sensitivity of a given layer to that type of perturbation, revealing its functional role. By aggregating these effects across prompt types and layers, we derive a structured view of how MLLMs process and integrate multimodal information.

We design each variant to target a distinct stage of the model's internal computation, enabling us to localize where different processing stages occur. **Lexical variants** test where the model begins aligning visual information with specific prompt phrasing—layers involved in grounding should be sensitive even to small wording changes. **Semantic negation** helps identify where the model begins to commit to an answer, as representations at these layers should reflect changes in the predicted outcome. However, because negation also changes the underlying visual concept being reasoned about, we introduce **output format variants** to decouple reasoning from decoding. These variants keep the visual input and its interpretation fixed while altering how the answer is expected to be expressed (e.g., "yes or no" vs. "1 or 0"), allowing us to test whether the model's internal representations encode the decision itself or just the tokens used to communicate it. Together, these controlled variations enable a fine-grained, layer-wise map of how multimodal information flows and transforms within the model.

Applying our framework to LLaVA-1.5 (Liu et al., 2023), we uncover a four-stage processing hierarchy: early layers (1–4) perform visual grounding, middle layers (5–13) integrate lexical information with visual features, and deeper layers transition into answer decoding. Notably, the output format variant further subdivides the middle-to-late stages, revealing that layers 12–15 are responsible for semantic reasoning—encoding the model's internal decision—while layers beyond 15 shift toward formatting that decision into specific output tokens. We find this structure to be preserved in LLaVA-Next-LLaMA-3 (Liu et al., 2024a), despite differences in visual tokenization, instruction tuning data, and the pretraining corpus of the base LLM—suggesting these factors do not fundamentally alter processing dynamics. In contrast, Qwen2-VL (Wang et al., 2024), which is built on a different LLM architecture (i.e, Qwen-LLM (Bai et al., 2023)), exhibits the same stage-wise structure but reallocates layer depth across stages, using fewer layers for grounding and more for reasoning. This highlights the critical role of LLM architecture in determining how and where multimodal

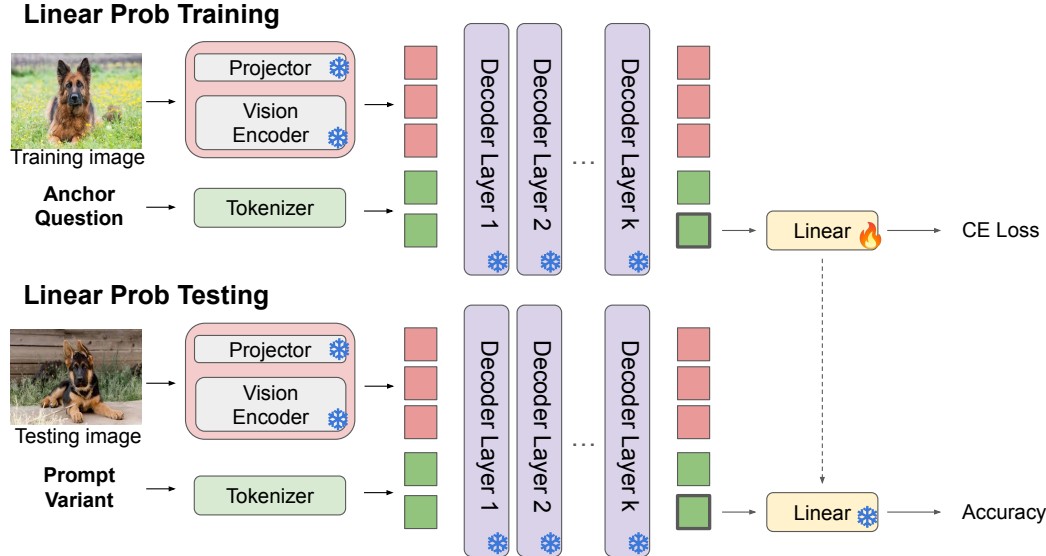

Figure 2: **Linear probing at decoder layer k.** Training (top): We pass training images with an anchor question through the frozen multimodal LLM, extract the sequence's last-token representation at layer k, and fit a linear probe on the ground-truth labels (in this example, for dog breed classification). Testing (bottom): we keep the probe fixed and evaluate it on last-token features from testing images under prompt variants (e.g., lexical rewrites, semantic negation), reporting classification accuracy. We repeat this for every decoder layer to obtain the layer-wise profile used in our analysis.

integration and decision formulation unfold within the model. This comparison is visualized in Figure 1, which highlights how different models distribute computational stages across depth.

## 2 Methodology

Our study investigates how multimodal large language models process information within the LLM across different layers by analyzing their token embeddings under varying prompt conditions. We employ linear probing on layer-wise embeddings to understand how representations evolve when the prompt question changes with constant visual inputs. An illustration of our probing framework can be found in Figure 2.

### 2.1 Experiment Setup

**Dataset and Anchor Question** We use a subset of ImageNet containing exclusively dog breed classes with the class-agnostic anchor question: *"Does this image show an animal? The answer must be always yes or no."* Before extracting embeddings, we first verify model compliance by running the anchor prompt through the MLLM for all images. We discard any images that do not receive a confident "yes" response. This filtering step removes cases where the model fails to follow instructions or misclassifies the image content, ensuring the model properly attends to the visual content and strictly maintains the "always yes" condition. The filtering process is applied when conducting evaluation with various prompt questions (details in Section 2.2), therefore, we move 300 images per class from the original ImageNet training split to the validation set. This allows us to retain enough examples even after discarding cases where the model fails to produce the expected answer.

We choose fine-grained visual classes (i.e., different dog breeds) rather than coarse categories to ensure that the classification task remains non-trivial. If the visual classes were too

coarse (e.g., "animal" vs. "non-animal"), the decision boundaries would be too robust to reveal small changes in internal representations caused by prompt variation, making probe sensitivity difficult to interpret. Additionally, we use a single anchor prompt shared across all training instances to avoid introducing prompt-textual cues that correlate with class labels. Using class-specific prompts would allow probes to memorize prompt features rather than genuinely reflect changes in how the model integrates visual and textual information.

**Embedding Extraction**    For the filtered dataset, we extract embeddings using each model's default prompt template. For example, LLaVA-1.5 uses the following structure:

```
USER: <image>
{Does this image show an animal? The answer must be always yes or no.}
ASSISTANT:
```

Other models employ their own standardized formats. For each model and image $x$, we collect the embedding $e_x^{(l)}$ of the last token at layer $l$, which captures the model-specific multimodal representation combining both visual and textual information at each processing stage.

**Probing Model Training**    We train linear classifiers $f^{(l)}$ at each layer to predict dog breeds $y \in \{1, \ldots, N\}$ from embeddings $e_x^{(l)}$, minimizing the cross-entropy loss:

$$\mathcal{L}^{(l)} = -\frac{1}{|\mathcal{B}|} \sum_{(x,y) \in \mathcal{B}} \log f_y^{(l)}(e_x^{(l)})$$

where $\mathcal{B}$ denotes a batch of training samples.

We train each probing model using the Adam optimizer with a learning rate of 0.001 and batch size of 16. Training is performed independently for each layer until convergence.

## 2.2    Evaluation Protocol

The core objective of our evaluation is to understand how different layers of multimodal LLMs encode and process information, using carefully designed prompt variations. After training probing models on embeddings generated from the standardized anchor question, we systematically evaluate them using three categories of modified prompts, each targeting a distinct aspect of the model's internal computation.

**Lexical variants** introduce minor surface-level changes (e.g., "image" → "picture") without altering the semantic meaning or expected answer. They help identify where the model begins aligning visual features to textual input: layers involved in this alignment should exhibit changes in representation—and thus drops in probing accuracy—even for subtle phrasing shifts. These variants localize the onset of visual-linguistic integration, showing where prompt wording begins to influence image-conditioned representations.

**Semantic negation variants** alter the core visual concept in the prompt (e.g., "animal" → "plane"), flipping the expected answer from "yes" to "no" while preserving syntactic structure. These variants help identify layers where the model begins encoding its internal decision: if representations are entangled with the answer label, they should diverge from the anchor despite identical images. A key challenge, however, is that semantic negation also changes the visual concept the model must reason about—introducing variation in the reasoning path and making it harder to isolate where answer commitment occurs.

To address this, we introduce the **output format variant**, in which the question remains fixed but the expected answer is altered in form only—for example, "yes/no" becomes "1/0." Since the model must arrive at the same semantic answer regardless of the output token, this variant allows us to decouple reasoning from decoding. If a model has already completed its reasoning at a given layer, the representation should remain consistent across output formats, and any changes should appear only in later decoding layers. Thus, this

variant isolates the transition point between reasoning about the input and preparing to generate the final answer.

Together, these variants allow us to characterize the full trajectory of multimodal processing—from early visual-text alignment, to semantic reasoning, to final answer generation. For each variant type, we evaluate probing accuracy layer-by-layer and compare it to the anchor baseline. We include only examples where the model produces the expected answer to both anchor and variant prompts, ensuring that observed differences reflect changes in internal representations rather than model failure.

## 3 Results

We begin by analyzing layer-wise processing patterns in LLaVA-1.5-Vicuna-7B (Liu et al., 2023) as a representative open-source multimodal LLM, and then extend this analysis to other popular models. All models evaluated in this section are of relatively modest scale (7B–8B parameters) due to limited compute resources.

### 3.1 Probing Analysis for LLaVA-1.5

Our probing framework trains layer-wise linear classifiers $f^{(l)}$ on last-token embeddings extracted using a standardized anchor question: *Does this image show an animal? The answer must be always yes or no.* (see Section 2). LLaVA-1.5 (Liu et al., 2023) uses Vicuna-7B (Chiang et al., 2023) as its base LLM, which consists of 32 transformer layers with an embedding dimension of 4096. Accordingly, we train 32 separate probing models, one for each layer, using the corresponding embeddings.

We first evaluate these probing models using embeddings generated from two categories of prompt variations: **Lexical Variants**, which modify the surface form of the anchor question while preserving its meaning, and **Semantic Variants**, which introduce semantic negation by altering the visual concept referenced in the prompt.

For lexical variants, we implement two specific edits: (1) substituting image-referencing nouns (e.g., *image → picture*), and (2) varying verbs (e.g., *show → feature*). For semantic variants, we replace the semantic keyword *animal* with an unrelated concept, e.g., *plane*, thereby changing the expected answer from *yes* to *no*. All evaluations are performed on embeddings from the same set of test images, after filtering to ensure that the model produces the expected responses for both anchor and variant prompts.

**Early and Middle Layers: Grounding and Integration** As shown in Figure 3, even at very early layers (1–2), a subtle but consistent increase in accuracy is observed for the anchor question alone (i.e., without prompt variation), suggesting the emergence of initial visual grounding. When lexical and semantic variants are introduced at test time, early layers (1–4) continue to exhibit relatively stable accuracy, indicating that these layers primarily encode visual information into the last-token embeddings without yet reflecting changes in the textual prompt. In contrast, middle layers (5–13) show substantial sensitivity to lexical changes; even minor modifications, such as altering the verb (*show → feature*), lead to an accuracy drop of approximately 40% at layer 13. These observations indicate a clear internal shift in LLaVA-1.5: while early layers (1–4) focus on visual grounding regardless of the prompt content, the alignment and integration of textual and visual information predominantly occurs in the middle layers (5–13).

**Late Layers: Reasoning vs. Answer Decoding** A critical behavioral transition occurs around layer 14, where the accuracy trajectories of lexical and semantic variants begin to diverge significantly. Specifically, accuracy for lexical variants recovers in deeper layers and becomes closely aligned with the anchor prompt, while accuracy for semantic variants remains consistently low. This contrast reflects a key difference in how the two variant types are constructed: lexical variants preserve the expected answer (*yes*), whereas semantic negation variants alter both the reasoning target (e.g., "animal" → "plane") and the expected answer (*no*). As a result, the persistent accuracy drop for semantic variants likely reflects not only sensitivity to conceptual shift but also commitment to a different output. This

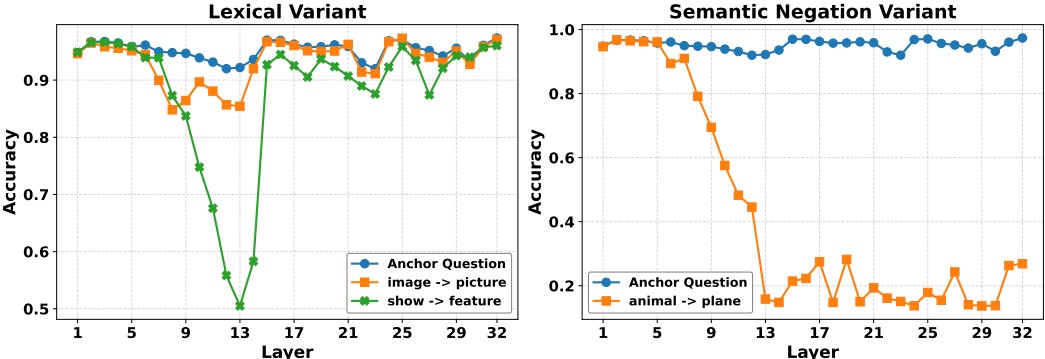

Figure 3: **Probing accuracy across layers for lexical and semantic negation variants.** Left: Lexical variants preserve the expected answer while modifying surface phrasing (e.g., `image` → `picture`, `show` → `feature`). Accuracy drops sharply in middle layers, revealing where visual-textual alignment occurs, but recovers in deeper layers as the model converges on the correct decision. Right: Semantic negation introduces a shift in both visual reasoning and expected answer (e.g., `animal` → `plane`), causing a persistent accuracy drop in deeper layers, indicating divergence in the model's internal decision process.

makes interpretation more challenging—probing accuracy alone cannot disentangle whether deeper layers reflect internal decision-making or the formatting of that decision into specific answer tokens. To resolve this ambiguity, we introduce output format variants in the next section, which modify only the expression of the answer while keeping the reasoning path and expected outcome fixed.

## 3.2 Decoupling Reasoning Paths from Answer Decoding

As observed in Section 3.1, deeper layers (around layer 14 and beyond) begin to diverge in how they represent prompts with different semantic content and expected answers. However, in the case of semantic variants (e.g., changing *animal* to *plane*), this divergence reflects two coupled changes: a shift in the model's internal reasoning target and a shift in the expected output. Specifically, switching from a positive (*yes*) prompt to a negative (*no*) one requires the model to realign visual features with a different semantic category, fundamentally altering its reasoning trajectory. As a result, the observed decline in probing accuracy for semantic variants cannot be attributed solely to output decoding but may also reflect deeper semantic reprocessing. To disentangle these factors, we introduce **output format variants**, which modify only the expected answer format (e.g., *yes/no → 1/0*) while keeping the reasoning path and semantic content identical to the anchor question.

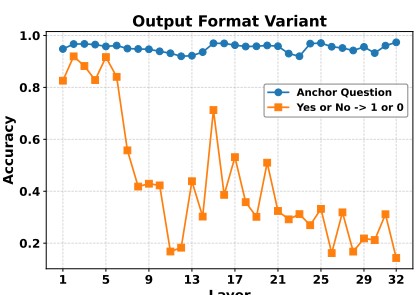

Figure 4: Probing accuracy when altering only the answer output format from textual (*yes/no*) to numerical (*1/0*).

This setup enables us to localize the transition point between semantic reasoning and output formatting: since the visual input and question remain fixed, any observed differences in internal representations must stem from how the model prepares to generate the answer in a specific output format, rather than from differences in the reasoning process itself.

Figure 4 shows the probing accuracy across layers under this controlled evaluation. Early layers (before layer 12) show accuracy patterns similar to those under lexical variations, reflecting sensitivity to minor wording changes in the prompt. Notably, accuracy under the output format variant rises from layers 12 to 15 before declining in deeper layers. We hypothesize that this trend highlights a separation between semantic reasoning and output

formatting: the initial increase occurs because the output variant and the anchor question share the same reasoning path, allowing the probe to generalize. The subsequent decline reflects divergence in answer format (*yes/no* vs. *1/0*), as deeper layers increasingly encode the specific form of the model's final response. These results help localize the transition from reasoning to decoding within the model's hierarchy.

**Summary.** Our layer-wise probing analysis reveals four distinct phases in LLaVA-1.5's multimodal processing pipeline:

- **Visual Grounding (Layers 1–4):** Early layers primarily encode visual input, producing representations that are largely invariant to both lexical and semantic prompt variations. This indicates that the model has not yet begun integrating language context into its processing.

- **Lexical Integration (Layers 5–11):** In this stage, representations become sensitive to surface-level phrasing changes in the prompt. This suggests that the model is aligning visual features with linguistic cues, marking the onset of multimodal interaction.

- **Semantic Reasoning (Layers 12–15):** These upper-middle layers reflect the model's internal decision-making process. Representations here diverge depending on the semantic content of the prompt, indicating that the model has committed to a specific interpretation of the input.

- **Answer Formatting (Layers 16+):** Late layers focus on shaping the final output. Representations are increasingly influenced by the desired response format rather than the input content itself, as confirmed by output format variant experiments.

## 3.3 Determinants of Multimodal Processing Mechanisms

Building on our layer-wise probing analysis of LLaVA-1.5 (Section 3.1), we now investigate which factors govern the internal processing behavior of multimodal LLMs. Contemporary MLLMs differ across three key dimensions: (1) *visual tokenization*—LLaVA-1.5 uses a fixed 576-token representation, while LLaVA-Next (Liu et al., 2024a) increases this by $4\times$ through multi-resolution chunking, and Qwen2-VL (Wang et al., 2024) adopts dynamic-resolution encoding (Dehghani et al., 2023); (2) *instruction tuning data*, which varies in both scale and content across models; and (3) *base LLM characteristics*, including both architectural design (e.g., LLaMA-variations (Touvron et al., 2023b; Chiang et al., 2023) vs. Qwen (Bai et al., 2023)) and pretraining corpus (e.g., Vicuna (Chiang et al., 2023) vs. LLaMA-3 (Dubey et al., 2024)). By comparing LLaVA-1.5-Vicuna-7B (Liu et al., 2023), LLaVA-Next-LLaMA-3-8B (Liu et al., 2024a), and Qwen2-VL-7B (Wang et al., 2024), we isolate which aspects of the processing hierarchy remain stable and which are sensitive to architectural or training choices.

### 3.3.1 Processing Dynamics Are Invariant to Tokenization and Training Data

To test whether components such as visual tokenization, instruction tuning data, or the base LLM's pretraining corpus fundamentally alter the processing behavior of multimodal LLMs, we compare **LLaVA-Next-LLaMA-3-8B** to **LLaVA-1.5** using our probing framework. These models differ along several key dimensions: LLaVA-Next adopts a *denser visual tokenization scheme* ($4\times$ more tokens via finer image chunking), is trained on a *larger and more diverse instruction tuning corpus*, and uses a base LLM (`LLaMA-3`) that shares architecture with LLaVA-1.5's Vicuna but differs substantially in its *pretraining data*.

As shown in Figure 5a, LLaVA-Next-LLaMA-3 exhibits probing trends that closely mirror those of LLaVA-1.5 (Figure 3) under both lexical and semantic variants: early layers show rising accuracy due to visual grounding, followed by a sharp drop from lexical variation in the middle layers. In deeper layers, accuracy for lexical variants that preserve the expected answer recovers and aligns with the anchor prompt, while semantic negation variants remain consistently low. Similarly, under output format variation (Figure 6a), LLaVA-Next-LLaMA-3 shows an accuracy peak from layers 10–15, corresponding to semantic reasoning—again matching the behavior observed in LLaVA-1.5. These results suggest that

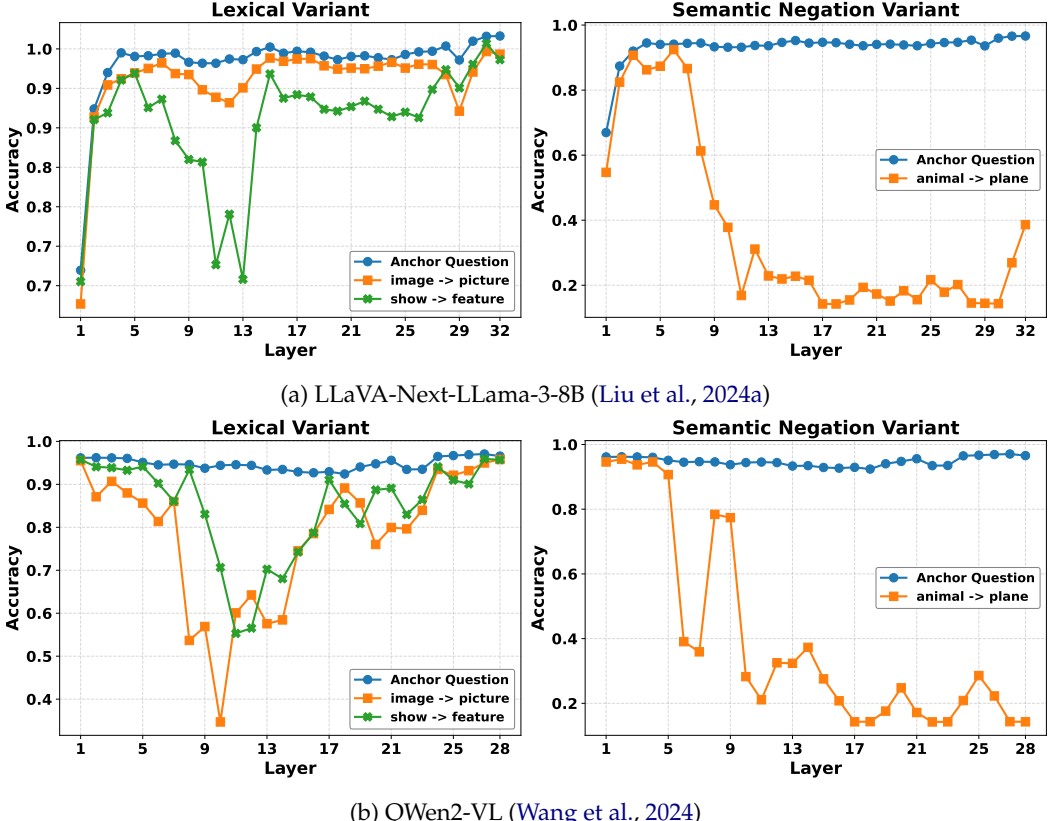

(a) LLaVA-Next-LLama-3-8B (Liu et al., 2024a)

(b) QWen2-VL (Wang et al., 2024)

Figure 5: Probing analysis with Lexical Variant and Semantic Negation Variant for LLaVA-Next-LLaMA-3-8B (Liu et al., 2024a) and Qwen2-VL (Wang et al., 2024). LLaVA-Next exhibits trends consistent with LLaVA-1.5, indicating that visual tokenization, instruction tuning data, and pretraining corpus have minimal impact on the overall processing hierarchy. Qwen2-VL shows a similar stage-wise trend but allocates fewer layers to visual grounding, reflecting differences in base LLM architecture.

processing dynamics are robust to changes in visual tokenization, instruction tuning data, and the base LLM's pretraining corpus, as long as the underlying architecture remains fixed.

### 3.3.2   *Base LLM Architecture Shifts Processing Depth, Not Structure*

While visual tokenization, instruction tuning data, and the base LLM's pretraining corpus do not substantially impact processing trends, the architecture of the base LLM itself does. To investigate this, we compare **Qwen2-VL** with **LLaVA-1.5** using our probing framework. Unlike LLaVA-1.5, which is built on Vicuna (a LLaMA-based architecture), Qwen2-VL is based on the Qwen family of LLMs—an entirely different transformer design, trained with its own tokenizer, rotary embedding scheme, and pretraining corpus.

As shown in Figure 5b, Qwen2-VL maintains high probing accuracy under lexical variants only in the first layer, with accuracy dropping sharply afterward and remaining low through layer 10. This indicates that integration between image and text tokens in Qwen2-VL begins as early as layer 2 and continues through layer 10. This range is broadly comparable to LLaVA, where lexical variant accuracy also declines across the first 11 layers. Thus, both models use a similar number of early layers for multimodal integration (visual grounding and lexical alignment).

Beyond the early layers, the two models begin to diverge more noticeably in how they allocate depth to later processing stages. Qwen2-VL extends the semantic reasoning stage over a longer range. As shown in Figure 6b, output format accuracy rises steadily from

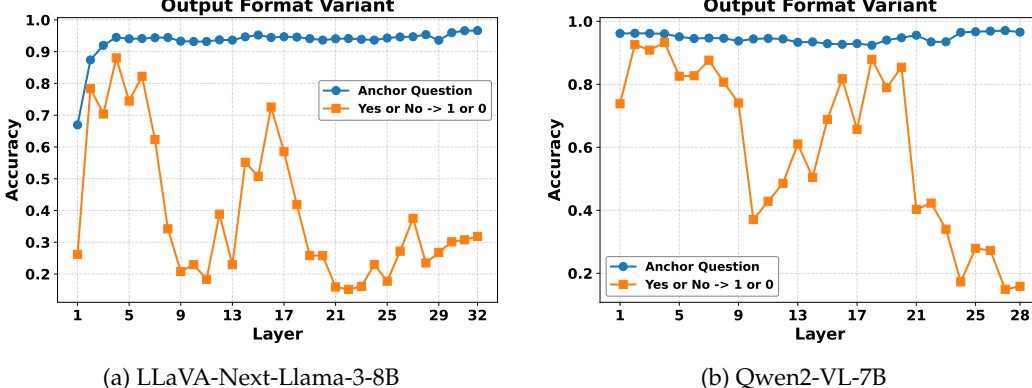

(a) LLaVA-Next-Llama-3-8B         (b) Qwen2-VL-7B

Figure 6: Probing accuracy on format-instruction variants (*yes/no* to *1/0*). Layer-wise accuracy initially increases (layers 12–15), highlighting semantic reasoning. Deeper layers show accuracy drops, indicating a transition to answer-decoding stages, especially prominent with stronger base LLMs like LLaMA-3. Qwen2-VL exhibits a similar trend but allocates a broader range of layers to reasoning, with a more gradual transition into decoding, reflecting architectural differences in how decision formulation and output formatting are handled.

layer 10 to 20, indicating a prolonged period during which internal representations are refined toward task-specific decisions. By contrast, LLaVA-1.5 exhibits a shorter reasoning window—typically confined to layers 12 to 15—after which decoding behavior dominates. This implies that Qwen2-VL leverages a deeper portion of its network for reasoning rather than output formatting, potentially enabling more flexible decision encoding before token generation.

In the final layers of Qwen2-VL (layer 21-28), the accuracy for lexical variant recovers to the level of anchor questions, whereas the accuracy under semantic negation remains low, indicating those layers are primarily responsible for answer decoding. By contrast, LLaVA allocates a larger portion of late layers for decoding, with lexical recovery appearing later and spanning more layers.

In summary, LLaVA and Qwen2-VL follow a common internal processing mechanism to solve image tasks—visual grounding, lexical integration, task reasoning and output decoding—yet the depth allocated to each stage and the transition points differ by model. In other words, the choice of different base LLM controls layer allocation across stages, not the internal pattern of processing.

## 4  Related Work

**Large Language Models** Large Language Models (LLMs) have demonstrated strong language understanding and generation capabilities, enabled by scaling transformer architectures and training on large text corpora (Brown et al., 2020; Touvron et al., 2023a; Zhang et al., 2022). Building on this foundation, recent work extends LLMs to vision-language tasks by integrating pretrained vision encoders with frozen or lightly finetuned language backbones (Liu et al., 2024b; Li et al., 2023; Zhu et al., 2023; Bai et al., 2023; Wang et al., 2024). These multimodal LLMs (MLLMs) align image features to the LLM input space via a projection or adapter, followed by instruction tuning on limited multimodal data. Rather than optimizing performance, our goal is to understand how MLLMs process visual and textual inputs internally. We introduce a layer-wise probing framework that analyzes token embeddings under controlled prompt variations, revealing a consistent stage-wise structure: early layers handle visual grounding, middle layers support reasoning, and later layers perform decoding—modulated by architectural choices.

**Interpretability of Large Language Models.** A growing body of work has developed techniques to interpret the internal mechanisms of large language models (LLMs), including causal tracing (Meng et al., 2022; Vig et al., 2020; Heimersheim & Nanda, 2024), logit

lens (Nanda, 2023), sparse autoencoders (Lan et al., 2024; He et al., 2024; Gao et al., 2024; Dunefsky et al., 2024), and probing (Hewitt & Manning, 2019; Wang et al., 2023; Yeo et al., 2024). These tools have recently been extended to multimodal LLMs (MLLMs), with causal tracing emerging as especially useful. For example, Basu et al. (2024) and Neo et al. (2024) apply it to LLaVA (Liu et al., 2024b), showing that early layers encode contextual cues, while middle layers focus on object-level features. Golovanevsky et al. (2024) compare cross-attention in BLIP-2 and LLaVA, finding that BLIP-2 performs both object detection and suppression, whereas LLaVA primarily suppresses outliers. Complementary to these efforts, Huo et al. (2024) use the logit lens to show that image embeddings often encode mixtures of semantic concepts rather than single-token alignments.

While existing interpretability methods provide valuable insights, they often require model-specific instrumentation or intervention. In contrast, we use linear probing as a lightweight and architecture-agnostic approach that operates directly on intermediate embeddings. By training layer-wise classifiers under controlled prompt variations, we identify when MLLMs perform visual grounding, reasoning, and decoding—without modifying model behavior or relying on custom tracing infrastructure.

## Limitations

Although our study targets the LLaVA family—in which a frozen vision encoder passes projected image tokens directly into the language model—multimodal architectures vary widely in how and where they blend modalities. Pre-LLM fusion systems such as the BLIP series (Li et al., 2023; Dai et al., 2023) insert a joint transformer before the decoder, while early-fusion designs like Chameleon (Chameleon Team, 2024) interleave image and text tokens from the first layer. These choices, together with differences in visual-token granularity, co-training versus freezing of the vision tower, and task objectives, can redistribute or blur the grounding, lexical integration, reasoning, and decoding stages we identify for LLaVA. Extending our layer-wise probing framework to these alternative designs—and verifying whether the same stage pattern still emerges—remains an important direction for future work.

## 5    Conclusion

We present a probing-based framework for analyzing the layer-wise processing dynamics of multimodal large language models. Applying our method to LLaVA-1.5, we identify a three-stage hierarchy: early layers (1–4) of the LLM perform visual grounding, middle layers (5–13) integrate lexical and visual information, and deeper layers transition into answer decoding. Output format variations further reveal that layers 12–15 encode semantic decisions, while layers beyond 15 shift toward formatting those decisions into specific output tokens. We observe this structure is preserved in LLaVA-Next-LLaMA-3, despite differences in visual tokenization, instruction tuning data, and pretraining corpus—indicating these factors do not substantially alter the model's internal processing flow. In contrast, Qwen2-VL reallocates layer depth across stages, with shallower grounding and extended reasoning, highlighting how base LLM architecture shapes multimodal integration across depth. We hope this analysis framework can serve as a foundation for future interpretability studies and facilitate more principled comparisons across multimodal model designs.

## Acknowledgment

This work was supported in part by NSF IIS2404180, Microsoft Accelerate Foundation Models Research Program, and Institute of Information & communications Technology Planning & Evaluation (IITP) grants funded by the Korea government (MSIT) (No. 2022-0-00871, Development of AI Autonomy and Knowledge Enhancement for AI Agent Collaboration) and (No. RS-2022-00187238, Development of Large Korean Language Model Technology for Efficient Pre-training).

We also thank Xueyan Zou for insightful discussions and helpful feedback.

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
