# OpenReview forum: "How Multimodal LLMs Solve Image Tasks: A Lens on Visual Grounding, Task Reasoning, and Answer Decoding"
_colmweb.org/COLM/2025/Conference — COLM 2025_

### Official Review · Reviewer_RU2Z · 2025-05-05

**Rating:** 7
**Confidence:** 3
**Ethics Flag:** 1

**Summary:**

This paper investigates how multimodal LLMs process visual and textual input across layers using a probing framework. To this end, linear classifiers are trained to predict fine-grained visual categories (dog breeds) from token embeddings extracted at each layer, using a class-agnostic prompt: “Does this image show an animal?” as an anchor question (answer is always “yes”) and baseline for the experiments. Three types of prompt variations are evaluated to investigate the role of different layers: lexical variants, semantic negation and output format variants. The results show a robust structure in MLLMs, with early layers performing visual grounding, middle layers supporting semantic reasoning and later layers preparing the task-specific output.

**Questions To Authors:**

See remarks above.

**Reasons To Accept:**

- the paper is clearly structured and well-written, the methodology is clearly explained.
- this research provides interesting insights in how visual and linguistic information interacts across layers in multimodal LLMs, using a straightforward approach that operates directly on intermediate embeddings. As such, this research is an interesting contribution to interpretability research for MLLMs.

**Reasons To Reject:**

- some concepts need a more detailed explanation/definition, e.g. “semantic reasoning”
- I understand the authors want to create a completely controlled experimental environment, but I would have preferred a more exhaustive set of experiments with different examples per  variant. If we take the lexical variants, for instance, you replace "image" > "picture", and "show" > "feature". As feature is also polysemous (even with distinct part-of-speech categories), it would be interesting to compare the results with other replacements where the verb is replace with another non-ambiguous verb. I think this study is very useful to give some first insights, but it could be expanded with a more elaborate test bed leading to deeper and more robust insights.
- It would be interesting to relate your results to findings in other interpretability research for LLMs, and more specifically, research investigating which information is encoded in the various layers. This is briefly mentioned in the related work section, but deserves a more elaborate discussion. This could better motivate why your approach, that uses linear probing as an architecture-agnostic approach,  is a more straightforward but viable method to investigate the inner workings of MLLMs.

---

> ### Author Response · Authors · 2025-06-02
> **Author Response**
>
> We are encouraged to see that the reviewer finds our work to provide interesting insights into how visual and linguistic information interact across layers in multimodal LLMs through a straightforward and interpretable approach. We appreciate the constructive suggestions and please see below for our response to your questions.
>
> **Q: Clarification on semantic reasoning**
>
> We agree that this term can benefit from more precise definition. In our context, *semantic reasoning* refers to the model’s ability to connect linguistic concepts in the prompt (e.g., the noun “animal”) with visual content (e.g., an image of a dog), and to derive a meaningful answer from that alignment.  We will clarify this along with other terminologies in the revised manuscript.
>
> **Q: A more exhaustive set of experiments with different examples per variant**
>
> We thank the reviewer for the insightful suggestion. We have explored additional lexical variations beyond those shown in the main paper, including “animal” → “pet”, “show” → “feature”, adding an extra question mark, and “image” → “picture”. These edits result in accuracy drops in descending order of severity, with more semantically impactful substitutions (e.g., “animal” → “pet”) leading to larger changes in probe performance.
>
> Crucially, these effects are concentrated in layers 5–13 of LLaVA-1.5, supporting our interpretation that these middle layers are responsible for integrating lexical signals with visual content. The consistent correlation between semantic strength of the lexical change and the degree of accuracy shift strengthens the conclusion that this layer range is sensitive to linguistic perturbations. We will include these expanded lexical results in the appendix of the next version of the manuscript.
>
> **Q: Relate your results to findings in other interpretability research for LLMs**
>
> We thank the reviewer for the thoughtful suggestion. We agree that better situating our work within the broader context of interpretability research for LLMs and MLLMs can clarify the motivation and significance of our approach.
>
> Our study offers a structured and fine-grained view of how multimodal large language models (MLLMs) process visual question answering tasks. Using prompt-based probing on the final token embedding across layers, we identify four distinct processing stages in LLaVA-1.5:
>
> 1. **Visual grounding** (layers 1–4)
> 2. **Lexical integration** (5–11)
> 3. **Semantic reasoning** (12–15)
> 4. **Decoding** (16+)
>
> This decomposition highlights functional specialization across the model's depth, revealing how different layers contribute to different stages of the multimodal pipeline.
>
> In contrast, prior works have not explicitly delineated these intermediate stages. For instance, **Basu et al.** analyze early layer MLPs and attention blocks to understand general information storage and transfer, but do not map how different linguistic perturbations affect the model’s reasoning flow. Similarly, **Neo et al.** track visual feature propagation in LLaVA but do not explore how linguistic variations interact with different processing stages or how semantic reasoning and output formatting emerge across layers.
>
> Our work complements these studies by introducing a controlled probing framework that isolates specific reasoning components through prompt variations. Importantly, our method is **model-agnostic and comparative**: we apply the same setup to **Qwen-VL** and **LLaVA-Next**, and observe how the layer-wise boundaries of each processing stage shift with architectural and training choices.
>
> We will expand the related work section to include this discussion and more clearly articulate how our approach builds on and extends prior interpretability efforts in both unimodal and multimodal LLMs.
>
> - Basu et al., *Understanding Information Storage and Transfer in Multi-modal Large Language Models*
> - Neo et al., *Towards Interpreting Visual Information Processing in Vision-Language Models*
>
> ---
> We sincerely thank the reviewer for the positive and constructive feedback. We're glad the contribution is seen as a valuable step toward MLLM interpretability, and we appreciate your suggestions to strengthen its framing. We would be grateful for your support during the discussion phase.

---

> > ### Comment · Reviewer_RU2Z · 2025-06-10
> >
> > Thank you for your answers. The proposed changes will improve the quality of the paper.

---

### Official Review · Reviewer_C29N · 2025-05-06

**Rating:** 5
**Confidence:** 4
**Ethics Flag:** 1

**Summary:**

This work investigates how MLLMs process visual and textual information across different layers using a probing-based approach. Specifically, the authors train linear classifiers on the hidden states of the final token at each layer to track how representations evolve under varying prompts, thereby uncovering the functional roles of individual layers.
Using the anchor question “Does this image show an animal? The answer must be always yes or no.”, they introduce three categories of prompt variants: (a) surface-level edits (e.g., “image” → “picture”), (b) semantic negation (e.g., “animal” → “plane”), and (c) output format changes (e.g., “yes or no” → “1 or 0”).
Their findings include: (a) middle layers (layers 5–13) are highly sensitive to surface-level changes; (b) late layers (after layer 14) show a significant drop in probe accuracy under semantic negation; and (c) a rebound in accuracy for semantic negation variants are observed in the final layers (30–32).

**Questions To Authors:**

1. Wouldn't it be more appropriate to train the probing classifier to answer the binary question “Is this an animal?” in line with the model's actual prompt, rather than predicting dog breeds, which introduces a mismatch between the probing task and the model’s intended behaviour?
2. Does replacing “animal” with a different noun like “plane” truly constitute semantic negation, or does it instead introduce semantic drift by altering the category being reasoned about making interpretation ambiguous?
3. Could the controlled environment oversimplify multimodal reasoning, limit generalisability, introduce artificial prompt constraints, and ambiguity in prompt interpretation?

**Reasons To Accept:**

1. The paper is clearly written.
2. The authors applied the analysis to multiple models (LLaVA-1.5, LLaVA-Next-LLaMA-3, and Qwen2-VL), thus offering valuable comparative insights in how multimodal information is processed.
3. A probing-based framework that reveals the internal processing dynamics of multimodal LLMs across layers.

**Reasons To Reject:**

1) In the experimental setup, it seems incongruent to train the probing classifier to predict dog breeds, as this effectively poses the question “What is this dog?”, whereas the actual prompt given to the model is “Does this image show an animal?”. This mismatch introduces a disconnect between the probing task and the model’s intended behaviour. A more appropriate approach might be to extract hidden states from both positive and negative examples and train the probe to directly answer the binary question “Is this an animal?”, thereby aligning the probe’s objective with the original task.

2) The probe is trained using the hidden states of the last token at each layer. However, information propagation in transformer models is not confined to the final token. Prior studies [1,2] have shown that in early layers, important information is often distributed across earlier tokens. As such, relying solely on the last-token representation may overlook critical dynamics in the processing flow, thereby raising concerns about the completeness and faithfulness of the conclusions drawn from the probe’s performance.

3) The experimental design involving output format variants (e.g., changing “yes or no” to “1 or 0”) appears problematic and may not robustly support the authors’ conclusions. Since the semantic content remains unchanged while only the output token format is altered, it is unclear whether the observed differences in probing accuracy genuinely reflect a transition from semantic reasoning to answer formatting. Without stronger justification or complementary analyses, this setting may not provide reliable evidence for localizing the decoding stage.
4) Semantic negation involves replacing “animal” with another noun like “plane.” This does not only flip the expected answer; it changes the semantic category being reasoned about, introducing semantic drift. This makes interpretation ambiguous.

[1] Dissecting Recall of Factual Associations in Auto-Regressive Language Models https://aclanthology.org/2023.emnlp-main.751.pdf
[2] A Mechanistic Interpretation of Arithmetic Reasoning in Language Models using Causal Mediation Analysis https://aclanthology.org/2023.emnlp-main.435.pdf

---

> ### Author Response · Authors · 2025-06-02
> **Author Response (1/3)**
>
> We thank the reviewer for acknowledging the strengths of our paper, particularly the clarity of presentation, the multi-model analysis, and the design of our probing-based framework. We also appreciate the constructive feedback.
>
> **Q: Wouldn't it be more appropriate to train the probing classifier to answer the binary question “Is this an animal?” in line with the model's actual prompt**
>
> We thank the reviewer for the thoughtful suggestion regarding aligning the probing task with the anchor prompt “Is this an animal?” using binary classification. We have explored similar setups early in the project and found important limitations that informed our current design.
>
> Below, we illustrate the reviewer’s proposed setup:
>
>
> | Component             | Description                                                    |
> |-----------------------|----------------------------------------------------------------|
> | **Anchor Question**   | “Is this an animal?”                                           |
> | **Training Images**   | Mixture of dog images (positive) and plane images (negative)  |
> | **Probing Labels**    | Binary: 0 (dog), 1 (plane)                                  |
> | **Lexical Variant**   | e.g., “Is this object an animal?”                              |
> | **Probing Objective** | Train a binary classifier on last-token embeddings             |
>
> In this binary classification setup, the probe easily achieves near-perfect accuracy across all layers when evaluated with the anchor prompt, as the separation between visual classes (e.g., dogs vs. planes) is broad and easily learnable.
>
> However, we observed that applying lexical variants (e.g., "image" → "object") leads to virtually no change in accuracy across layers:
>
> | Prompt Type      | Layer 7 | Layer 8 | Layer 9 | Layer 10 | Layer 11 | Layer 12 |
> |------------------|---------|---------|---------|----------|----------|----------|
> | Anchor Prompt    | 99.7    | 99.9    | 100.0    | 100.0     | 99.9     | 100.0     |
> | Lexical Variant  | 99.7   | 99.8    | 99.9    | 100.0     | 99.7     | 99.9     |
>
> This outcome occurs because:
>
> - **The binary decision boundary between dog and plane is overly coarse.** These two categories are easily separable in the embedding space, leading to trivial classification.
>
> - **Lexical changes cause only small shifts in the representation space, which are insufficient to move embeddings across such a wide separation.** As a result, the probe maintains high confidence and accuracy across layers—even when the prompt is perturbed—making it ineffective for measuring sensitivity to prompt variations.
>
> - **Using binary labels for fine-grained classes (e.g., “Labrador” vs. “Beagle”) does not resolve the issue.** We explored this option by using prompts like “Is this a Labrador?” for each class; however, we found that models like LLaVA often default to answering “yes” for all dog images, regardless of the specific breed of the image. This makes the supervision signal unreliable and prevents the probe from learning a meaningful decision boundary.
>
> In contrast, our setup uses fine-grained visual classes (e.g., different dog breeds) and a multi-class classification probe. Since these classes are much closer together in the embedding space, even small changes in how the prompt is phrased can shift the final token representation enough to cross class boundaries and result in a drop in classification accuracy. This makes the probe a much more sensitive instrument for detecting how prompt perturbations affect internal representations. Our design thus enables clear attribution of variation sensitivity to specific layers, which would be obscured in the coarse binary classification setting.
>
> We appreciate the suggestion and hope this clarifies the motivation behind our experimental design. We will include a discussion of this experiment and comparison in the next version of the manuscript.
>
> **Q: Could the controlled environment oversimplify multimodal reasoning**
>
> As the first work to apply probing-based analysis to MLLMs, our goal is to establish a controlled and interpretable experimental foundation. By fixing visual inputs and systematically introducing prompt variations, we demonstrate that probing can be a powerful tool for understanding how multimodal information is processed across layers.
>
> The tasks are carefully designed to isolate specific processing components—such as grounding, semantic integration, and answer formatting—without confounding factors. We believe this level of control is essential for building a precise understanding of internal mechanisms before extending the methodology to more complex and realistic scenarios.

---

> > ### Author Response · Authors · 2025-06-02
> > **Author Response (2/3)**
> >
> > **Q: Why probing on last token embedding?**
> >
> > We thank the reviewer for raising this important point. While we recognize that early-layer information in transformer models can be distributed across multiple tokens, we intentionally probe the last token because it is the position from which autoregressive models generate the final answer—making it a consistent and interpretable target across layers.
> >
> > In multimodal LLMs like LLaVA, alternative strategies such as average pooling across all tokens are problematic. These models typically process 576 visual tokens, which far outnumber the number of textual tokens in most prompts. As a result, average pooling leads to representations that are dominated by visual content and largely insensitive to changes in the prompt. Additionally, because LLaVA uses an autoregressive decoder architecture, **the visual tokens appear before the textual prompt and cannot attend to it**. Since the model ultimately generates answers based on the final textual token, probing visual tokens would not reveal how the model internally processes the question or constructs an answer.
> >
> > By focusing on the final token under controlled prompt variations, we can isolate how internal representations evolve across layers in response to lexical, semantic, and output format changes. This allows us to uncover the functional roles of different layers (e.g., grounding, reasoning, decoding), which would be obscured under pooling-based alternatives.
> > We appreciate the reviewer’s suggestion and will clarify this design rationale in the final manuscript.
> >
> > **Q: Semantic negation involves replacing “animal” with another noun like “plane.” This does not only flip the expected answer**
> >
> > We respectfully disagree and would like to clarify the purpose and design of our semantic negation experiment.
> >
> > In our setup, **all training and testing images are of dogs from different breeds**, and the anchor question is *“Does this image show an animal?”* For semantic negation, we modify the prompt to *“Does this image show a plane?”*, which **flips the expected answer from “yes” to “no.”** We manually verified and filtered both training and test sets to ensure the model returns the correct answer under these conditions, validating that the label shift is meaningful and controlled.
> >
> > We fully acknowledge that replacing *“animal”* with *“plane”* alters the semantic category being reasoned about, and agree this introduces a confound: it becomes unclear whether observed accuracy drops reflect a change in **semantic commitment** (i.e., the answer the model intends to give) or a shift in the **reasoning target** required to get there.
> >
> > **This ambiguity is precisely what motivates the introduction of our output format variant.** By keeping the anchor question fixed and only modifying the form of the expected answer (e.g., *“yes or no”* → *“1 or 0”*), we isolate the decoding process from the reasoning process. **Together, the semantic negation and output format experiments form a controlled pair**—one manipulates the reasoning target while holding the answer format fixed, and the other manipulates the format while holding the reasoning target fixed. This design allows us to decouple the two effects and more accurately identify which layers are responsible for semantic reasoning versus answer generation.

---

> > > ### Author Response · Authors · 2025-06-02
> > > **Author Response (3/3)**
> > >
> > > **Q: The experimental design involving output format variants (e.g., changing “yes or no” to “1 or 0”) appears problematic and may not robustly support the authors’ conclusions.**
> > >
> > > We appreciate the reviewer’s concern and would like to clarify the motivation behind the output format variant. It is specifically designed to **decouple reasoning from decoding**—that is, to separate the model’s ability to arrive at the correct conceptual answer from its ability to express that answer in the required textual format.
> > >
> > > In this setting, both the **image** and the **question** (e.g., “Does this image show an animal?”) are fixed, and only the **expected answer format** is changed (e.g., from “yes or no” to “1 or 0”). Since the reasoning path is the same, the model’s internal conceptual answer—an affirmative in this case—should remain unchanged. What differs is how that internal answer is mapped to output tokens: in one case to “yes,” in another to “1.”
> > >
> > > This expectation is reflected in the results: accuracy remains stable through early and middle layers, indicating shared internal representations. However, we observe a drop in the final layers, suggesting they are responsible for converting the conceptual answer into the required surface form. The output format variant thus allows us to isolate this decoding behavior and provides complementary evidence for our interpretation of layer-wise processing stages in MLLMs.
> > >
> > > ---
> > > We thank the reviewer for the thoughtful feedback and engaging questions. We hope our responses have clarified the design, motivation, and interpretation of our analysis. We would be happy to provide further details if needed, and kindly ask you to reconsider your score in light of these clarifications.

---

> > ### Comment · Reviewer_C29N · 2025-06-10
> >
> > The authors provided a good explanation to my queries. I am happy to raise my evaluation by one point.

---

> > > ### Author Response · Authors · 2025-06-10
> > > **Regarding the evaluation updates**
> > >
> > > Dear Reviewer,
> > >
> > > Thank you again for your time and effort on providing constructive feedback for our work. We just want to kindly follow up, as I noticed the score hasn’t been updated yet (stilling showing 5 on our end).
> > >
> > > Thanks again for your time and feedback!

---

### Official Review · Reviewer_nNBQ · 2025-05-16

**Rating:** 4
**Confidence:** 4
**Ethics Flag:** 1

**Summary:**

This paper introduce a probing framework to systematically analyze how MLLMs process visual and textual inputs across layers.
Specificaly, the authors trained probing linear network on the last token of each layer of the LLM in MLLMs when answering visual classification questions. Then the author introduce several input variants of MLLMs and test the probing network on these variants and ues the classification accuracy drop to show the discrepency of LLM embeddings when changeing the MLLM input.

By systemmatical experiment, the authors find a consistent stage-wise structure in early layers perform visual grounding, middle layers support semantic reasoning, and later layers prepare task-specific outputs. The authors further show that such pattern is robust to training data and visual tokenization but shift with changes of base LLM.

**Questions To Authors:**

Figure 3 is redundant with Figure 5, it is suggested to carefully check the paper before submission.

In Figure 2, the image -> picture shows much less accuracy drop comparing with show -> feature, I'm wondering if the authors have any insights about this.

**Reasons To Accept:**

The experiment is well-designed and provide insightful interpretation to how MLLMs interact with multimodal information.

This paper also provide a useful protocol for future research when people want to understand MLLM behavior in more models and more tasks.

**Reasons To Reject:**

Some experiment conclusions need to be further justified.
For example, the authors claim that
> When lexical and semantic variants are introduced at test time, early layers (1–4) continue to exhibit relatively stable accuracy, indicating that these layers primarily encode visual information into the last-token embeddings without yet reflecting changes in the textual prompt.

It could be better if image variation is also included and see if the performance drop significantly when image is changed.

Only several simple visual question is studied, it's possible that the pattern observed in this paper might shift significantly when the model is answering a different type of visual question, such as OCR.

The writing needs to be substantially improved and the paper needs better organization, for example section 2.2 only mentioned Lexical variant, Semantic negation variants and output format variant, prompt structure variants is never mentioned. In addition prompt structure variants is introduced two times in section 3 (line 191 and 249).

---

> ### Author Response · Authors · 2025-06-02
> **Author Response**
>
> We thank the reviewer for their thoughtful and constructive feedback. We appreciate the recognition of our probing framework and its potential utility for future research. Below we address the concerns raised regarding experiment justification, scope, and presentation.
>
> **Q: What about including image variants?**
>
> We appreciate the reviewer’s suggestion and would like to clarify the motivation behind our experimental setup. The primary goal of this work is to understand how multimodal large language models (MLLMs) solve image-based tasks internally, and in particular, to dissect the distinct roles that different layers play in this process.
>
> To achieve this, we train probing models using the **training split** of a dog-breed subset from ImageNet and evaluate on its corresponding **test split**, which naturally introduces variation in image content between training and testing. During evaluation, we apply multiple **prompt variants** to the **same test set**: one with the original **anchor prompt** and others with controlled variations—such as lexical substitutions, semantic negations, or output format changes. The probing model is trained only with the anchor prompt on the training split, and remains fixed during testing. By comparing accuracy across these prompt conditions on the same set of test images, we can isolate the model’s sensitivity to different types of linguistic change. This design enables us to analyze how different layers contribute to key stages of multimodal processing, including visual grounding, lexical integration, reasoning, and decoding.
>
> **Q: Only several simple visual question is studied**
>
> We appreciate the reviewer’s suggestion and agree that probing models under different types of visual questions (e.g., OCR) is an interesting direction for future work. However, our decision to focus on simple class-discriminative questions (e.g., “Does this image show an animal?”) is intentional and driven by the design constraints of probing.
>
> Specifically, probing requires a stable, categorical target for supervision. In our setup, we use an image classification dataset (ImageNet dog breeds), where each image has a clearly defined class label, enabling us to train probing models that map internal embeddings to consistent targets. For tasks like OCR, however, the expected output is typically open-ended or continuous (e.g., a word or phrase), and it becomes unclear how to define class labels suitable for probing. Constructing a probing task would require grouping images by identical OCR text content—effectively treating each string as a class—which is impractical given the open vocabulary and long-tail distribution of textual content in natural images.
>
> Moreover, our objective is to understand how different layers of an MLLM contribute to the internal processing pipeline, rather than to benchmark performance on diverse VQA tasks. Using simple visual questions allows us to control the linguistic input precisely, isolate the effects of prompt variation, and cleanly attribute changes in accuracy to distinct processing stages such as visual grounding, lexical integration, and reasoning.
>
> **Q:Writing Organization and Redundant Figures**
>
> We thank the reviewer for pointing out the structural and organizational issues. We will revise Section 2.2 to introduce prompt structure variants alongside the other variant types for clarity and consistency. Additionally, we will remove the redundant Figure 5 and consolidate relevant insights to avoid repetition. These edits will be incorporated in the next version of the manuscript.
>
> **Q: In Figure 2, the image -> picture shows much less accuracy drop comparing with show -> feature**
>
> We thank the reviewer for the insightful question. We have explored additional lexical variations, including “animal” → “pet”, “show” → “feature”, adding an extra question mark, and “image” → “picture”. These yield accuracy drops in descending order of severity, with more semantically meaningful changes (e.g., “animal” → “pet”) producing larger drops.
>
> Crucially, these effects are concentrated in layers 5–13 of LLaVA-1.5, consistent with our interpretation that these middle layers are responsible for lexical-visual integration. The fact that the magnitude of the accuracy drop scales with the semantic strength of the change further supports the conclusion that these layers are actively sensitive to linguistic content. We will include this additional analysis in the appendix of the next version of the manuscript.
>
> ---
> We again thank the reviewer for the constructive feedback and insightful questions. We hope our clarifications have addressed your concerns and helped illustrate the rationale behind our experimental design. We are happy to provide any additional details if needed and kindly hope you to consider revisiting your score in light of these clarifications.

---

### Official Review · Reviewer_oi97 · 2025-05-24

**Rating:** 6
**Confidence:** 4
**Ethics Flag:** 1

**Summary:**

This paper investigates how recent multimodal large language models (MLLMs) process information across their layers, aiming to determine whether there is a hierarchical structure in how these models handle different types of information. The main objective of this work is to build a probing classifier for each layer's representations to assess whether MLLM layers are robust to certain properities such as visual grounding, lexical-visual integration, and task-specific information processing. To perform this, the authors consider three popular MLLMs LLaVA-1.5, LLaVA-Next-LLaMA-3, and Qwen2-VL. The results show that across all MLLMs: early layers encode visual grounding, middle layers support semantic reasoning, while later layers focus on task-specific outputs.

**Questions To Authors:**

The authors could explain why they chose only two structured prompt variants for lexical changes (Image->Picture, Show->Feature), but only one prompt variant for semantic negation (Animal->plane). It is unclear how these specific variants were selected and what criteria were used to define them as representative. The paper would benefit from either referencing prior literature that supports the use of these variants or explaining whether alternative variants were considered. Additionally, it would be helpful to know if the authors explored other commonly used approaches from past studies that examine model robustness through diverse prompt structures.

**Reasons To Accept:**

1. Probing MLLMs is a valuable direction, especially given the growing popularity of instruction-tuned MLLMs for zero-shot and general-purpose tasks. Understanding how information is processed at each layer can offer deeper insights into the hierarchical structure of these models.
2. The authors evaluate multiple MLLMs, and the consistency of results across models suggests that the findings may generalize well beyond a single architecture.
3. Exploring different prompt structure variants adds to the robustness of the analysis, showing how these models handle variations in input format and maintain stable processing across prompts.

**Reasons To Reject:**

1. Unlike previous probing studies in language models, such as [Conneau et al. 2018],[Tenney et al. 2018],[Jawahar et al. 2019], which use a wide range of probing tasks to understand the type of information encoded at each layer, this study relies on only three probing tasks. As a result, it does not offer a comprehensive view of how information is processed across layers in MLLMs. Additionally, the use of just three prompt variants limits the strength of the conclusion that these models follow a hierarchical structure of information processing. Incorporating a broader set of probing tasks-similar to those used in earlier language model studies-would provide a more thorough and interpretable analysis of these models.


[Conneau et al. 2018] What you can cram into a single \$ \&!\#* vector: Probing sentence embeddings for linguistic properties, ACL-2018

[Tenney et al. 2018] What do you learn from context? Probing for sentence structure in contextualized word representations, ICLR-2018

[Jawahar et al. 2019] What does BERT learn about the structure of language?, ACL-2019

2. The Limited structure variations in input prompts (Image->Picture, Animal->plane) are not sufficient to generalize claims about how these models process information. Also, if an MLLM has been pretrained on datasets like ImageNet, it is possible that the some of the observed robustness inherits from prior knwowledge rather than true generalization. Further, in typical probing studies, tasks are designed to be out-of-domain so that the model has no direct knowledge of the tasks themselves. Each probing task is meant to isolate and test specific properties, ensuring that the results reflect what the model has truly learned, rather than what it has memorized.

3. The results presented in the paper are more observational based on Figs, lacking deeper implications or strong conclusions drawn from each finding.

---

> ### Author Response · Authors · 2025-06-02
> **Author Response (1/2)**
>
> We sincerely thank the reviewer for recognizing the value of our work in probing multimodal large language models (MLLMs), highlighting the generalizability of our findings across multiple model architectures, and appreciating the robustness contributed by our prompt variations. We address the specific comments and questions below.
>
> **Q: This study relies on only three probing tasks.**
>
> We respectfully disagree that the number of probing tasks is a significant weakness of our study. We believe that the strength and clarity of experimental design lie **not in the quantity of tasks, but rather in their appropriateness for clearly conveying the intended insights**. Specifically, our experimental setup was carefully tailored to address our core research question: *how do multimodal large language models (MLLMs) internally process information when performing visual tasks?*
>
> Our probing setup follows a controlled strategy: we train a linear classifier using embeddings from a fixed anchor prompt and then evaluate the same classifier on prompt variants at test time. These variants introduce targeted linguistic changes—such as lexical substitutions, semantic negation, or format reformulations—while keeping the visual input constant. The drop in classification accuracy between anchor and variant reveals which layers are sensitive to each type of change. For example, lexical variants modify surface wording without altering meaning; layers that show large performance drops in this case are likely encoding lexical form. By analyzing these shifts across layers and variant types, we are able to clearly identify distinct processing stages within the model, including visual grounding, lexical integration, semantic reasoning, and decoding.
>
> We acknowledge and fully respect that prior linguistic probing studies (e.g., Conneau et al., 2018; Tenney et al., 2018; Jawahar et al., 2019) employed a broader range of evaluation settings. However, their objective was fundamentally different: to analyze the fine-grained linguistic structure learned by unimodal language models. Such work naturally requires diverse linguistic phenomena and syntactic constructions. In contrast, our focus is on uncovering how multimodal models organize and integrate visual and textual information internally. Our controlled probing setup, combined with carefully designed prompt variants, enables us to isolate specific representational behaviors with clarity. This compact yet diagnostic approach is well suited to our goal of identifying distinct stages of multimodal processing.
>
> **Q: it is possible that the some of the observed robustness inherits from prior knowledge rather than true generalization**
>
> We would like to clarify that our method is **not intended to evaluate what the model has “truly learned” in the sense of traditional NLP probing studies**, which typically use out-of-domain tasks to avoid overlap with pretraining data. Prior works such as Conneau et al. (2018), Tenney et al. (2018), and Jawahar et al. (2019) aim to test whether internal representations encode linguistic abstractions—like syntax, sentence structure, or semantics—by using a broad suite of carefully designed linguistic tasks. This broad task coverage is necessary given the complexity and diversity of linguistic phenomena.
>
> In contrast, our objective is to understand how different layers of a multimodal large language model (MLLM) operate during the inference process of visual question answering. We do not rely on distributional shifts or task novelty. Instead, our focus is on measuring relative sensitivity across layers using a consistent probing setup: training linear probes on last-token embeddings from a fixed anchor prompt, and testing on controlled prompt variants. Since both anchor and variant operate within the same visual input and prompt structure, our analysis isolates internal behavioral shifts—revealing which layers respond to surface-level, semantic, or structural changes. The insight comes from these controlled comparisons, not from testing generalization beyond pretraining data.

---

> > ### Author Response · Authors · 2025-06-02
> > **Author Response (2/2)**
> >
> > **Q: why two structured prompt variants for lexical changes (Image->Picture, Show->Feature), but only one prompt variant for semantic negation (Animal->plane).**
> >
> > We would like to clarify that the specific prompt variants shown in the paper—such as “image”→“picture” and “show”→“feature” for lexical changes, and “animal”→“plane” for semantic negation—are illustrative examples within broader categories. While these variants were selected for clarity, we have explored a broader set of lexical modifications and consistently observed similar trends. These include adding dummy tokens (e.g., “Does this image, perhaps, show an animal?”), redundant modifiers (“this particular image”), extraneous punctuation (“Does this image show an animal??”), and minor rephrasings (e.g., “Is there an animal in this image?”; see Fig. 5, left). **All of these lexical changes lead to accuracy drops concentrated in the middle layers, reinforcing our interpretation that these layers are responsible for lexical-visual integration**.
> >
> > Likewise, we also explored other options for the answer format variant beyond the “yes/no” to “1/0” substitution shown in the paper. For example, we tested alternatives such as “correct/incorrect,” “true/false,” and “positive/negative.” We found that LLaVA models are particularly brittle when handling these variants—often failing to produce consistent completions—whereas Qwen-VL is more robust. Despite these differences, both models exhibit similar trends in accuracy shifts at the final layers, further confirming the sensitivity of decoding stages to answer formatting.
> >
> > Due to rebuttal constraints, we are unable to revise the main text, but we will include these additional examples and supporting analysis in the appendix to further substantiate the robustness of our findings.
> >
> > **Q: Deeper implication of the results and comparison to prior work**
> >
> > Our study provides a comprehensive and nuanced understanding of how multimodal large language models (MLLMs) process visual question answering tasks. By employing controlled prompt variations and layer-wise probing, we identify four distinct processing stages in LLaVA-1.5:
> >
> > 1. **Visual grounding** (layers 1–4)
> > 2. **Lexical integration** (layers 5–11)
> > 3. **Semantic reasoning** (layers 12–15)
> > 4. **Decoding** (layers 16+)
> >
> > This structured decomposition offers a detailed view of the model's internal mechanisms, highlighting how different layers contribute to specific aspects of multimodal processing.
> >
> > In contrast, prior works have offered broader analyses without such granular differentiation. For instance, **Basu et al.** utilize attention-based analysis to explore information storage and transfer in MLLMs, focusing on the roles of MLP and self-attention blocks in early layers. Their study emphasizes the importance of these components in information storage but does not delineate distinct processing stages like lexical integration or semantic reasoning.
> >
> > **Neo et al.** investigate how visual information propagates through the language model component of LLaVA, analyzing the localization and evolution of object-relevant features. While this provides insight into visual token dynamics, it does not map out the full processing sequence from grounding to decoding, nor does it address how models respond to structured linguistic variations.
> >
> > Importantly, our method is not limited to a single model. In addition to LLaVA-1.5, we apply the same analysis to **Qwen-VL**, and observe differences in how processing stages are distributed across layers. These variations reflect architectural and training differences, and our framework offers a systematic way to compare how such design choices influence the functional specialization of layers in MLLMs.
> >
> > - Basu et al., *Understanding Information Storage and Transfer in Multi-modal Large Language Models*
> > - Neo et al., *Towards Interpreting Visual Information Processing in Vision-Language Models*
> >
> >
> > ---
> > We sincerely thank the reviewer for the thoughtful feedback and the opportunity to clarify our contributions. We hope our responses have addressed the concerns and provided a clearer understanding of our methodology and findings. We would be happy to provide further clarification if needed and kindly ask you to consider updating your score in light of these clarifications.

---

> > > ### Comment · Reviewer_oi97 · 2025-06-10
> > >
> > > I thank the authors for the clarifications. In particular, author explained how is the current work different from early probing studies in language models. I have updated my score accordingly.

---

> > > > ### Author Response · Authors · 2025-06-10
> > > > **Thank you for your response**
> > > >
> > > > We sincerely thank the reviewer for the time and effort invested in evaluating our work and we appreciate your constructive feedback and are glad that our revisions have addressed your concerns. We will refine our description of the probing framework and add additional results as described in our response in the next version of our manuscript.

---

### Decision · Program_Chairs · 2025-07-08

**Decision:**

Accept

**Comment:**

This paper presents a thoughtful study on the internal workings of Multimodal LLMs using a novel probing framework. I find the research question timely and the approach clever. The reviewers raised several valid points during the initial review phase. They questioned the limited scope and specific design choices, which could have weakened the paper's conclusions. These initial concerns were reflected in a wide spread of scores, indicating a clear need for clarification from the authors.

The authors' rebuttal was compelling and directly addressed the main criticisms. I was particularly convinced by their argument that their goal required a different, more controlled probing setup than prior work in unimodal NLP. Their careful explanations satisfied most reviewers, leading to two score increases (C29N was not able to update the score but verbally confirmed) and a strong consensus for acceptance among those who engaged in the discussion. While one reviewer remained negative and did not respond, their primary concerns were substantially resolved by the authors' clarifications. In my judgment, the paper makes a solid contribution to MLLM interpretability.

The AC considered the review, the author response and the discussion and decided to recommend Accept.